# Changes Induced by P2X7 Receptor Stimulation of Human Glioblastoma Stem Cells in the Proteome of Extracellular Vesicles Isolated from Their Secretome

**DOI:** 10.3390/cells13070571

**Published:** 2024-03-25

**Authors:** Fabrizio Di Giuseppe, Lucia Ricci-Vitiani, Roberto Pallini, Roberta Di Pietro, Patrizia Di Iorio, Giuliano Ascani, Renata Ciccarelli, Stefania Angelucci

**Affiliations:** 1Department of Innovative Technologies in Medicine and Dentistry, ‘G. d’Annunzio’ University of Chieti-Pescara, Via Vestini 31, 66100 Chieti, Italy; f.digiuseppe@unich.it; 2Center for Advanced Studies and Technology (CAST), ‘G d’Annunzio’ University of Chieti-Pescara, Via L Polacchi 13, 66100 Chieti, Italy; renata.ciccarelli51@gmail.com; 3Stem TeCh Group, Via L Polacchi 13, 66100 Chieti, Italy; 4Department of Medical, Oral and Biotechnological Sciences, ‘G d’Annunzio’ University of Chieti-Pescara, Via Vestini 31, 66100 Chieti, Italy; lucia.riccivitiani@iss.it; 5Department of Oncology and Molecular Medicine, Istituto Superiore di Sanità, Via Regina Elena 299, 00161 Rome, Italy; roberto.pallini@unicatt.it; 6Institute of Neurosurgery, Università Cattolica del Sacro Cuore, Largo Agostino Gemelli 8, 00168 Rome, Italy; roberta.dipietro@unich.it; 7Department of Medicine and Aging Sciences, ‘G. d’Annunzio’ University of Chieti-Pescara, Via Vestini 31, 66100 Chieti, Italy; patrizia.diiorio@unich.it; 8UOSD Maxillofacial Surgery, Azienda Sanitaria Locale di Pescara, Via Renato Paolini 47, 65124 Pescara, Italy; giuliano.ascani@asl.pe.it

**Keywords:** glioblastoma-derived stem-like cells (GSCs), microvesicles, exosomes, proteomic analysis, P2X7 receptors

## Abstract

Extracellular vesicles (EVs) are secreted from many tumors, including glioblastoma multiforme (GBM), the most common and lethal brain tumor in adults, which shows high resistance to current therapies and poor patient prognosis. Given the high relevance of the information provided by cancer cell secretome, we performed a proteomic analysis of microvesicles (MVs) and exosomes (EXOs) released from GBM-derived stem cells (GSCs). The latter, obtained from the brain of GBM patients, expressed P2X7 receptors (P2X7Rs), which positively correlate with GBM growth and invasiveness. P2X7R stimulation of GSCs caused significant changes in the EV content, mostly ex novo inducing or upregulating the expression of proteins related to cytoskeleton reorganization, cell motility/spreading, energy supply, protection against oxidative stress, chromatin remodeling, and transcriptional regulation. Most of the induced/upregulated proteins have already been identified as GBM diagnostic/prognostic factors, while others have only been reported in peripheral tumors. Our findings indicate that P2X7R stimulation enhances the transport and, therefore, possible intercellular exchange of GBM aggressiveness-increasing proteins by GSC-derived EVs. Thus, P2X7Rs could be considered a new druggable target of human GBM, although these data need to be confirmed in larger experimental sets.

## 1. Introduction

Glioblastoma multiforme (GBM) is the most aggressive brain tumor in adulthood, with high recurrence after current therapies and a poor patient survival rate [1]. Many factors account for the malignancy of GBM. Among them, a GSC niche within the tumor mass contributes to tumor growth, metastasis, increased resistance to therapy, and tumor relapse. Aligning with other cancer stem cells, GSCs are endowed with a high degree of proliferation/self-renewal and the ability to differentiate into multiple cells in the primary tumor, invade, and resist anti-cancer therapies, all accounting for tumor onset and growth [2].

An additional feature of GBM is the presence of EVs in the tumor microenvironment (TME) [3]. EVs are particles released from every type of cell and classified as exosomes (EXOs), microvesicles (MVs), and apoptotic vesicles. EXOs show a size ranging from 30 to 150 nm and are of endosomal origin, MVs (50–1000 nm) originate from plasma membranes, while apoptotic bodies (1–5 μm size) are generated during the process of cell death by apoptosis [4]. The pleiotropic role of these particles, mainly MVs and EXOs, in intercellular communications is due to their content, composed of many bioactive molecules, including proteins as well as lipids, nutrients, and nucleic acids, which can potentially be acquired by surrounding cells, thus affecting their function [4]. EVs are produced by cells under physiological and pathological conditions, including tumors. Thus, EVs are released from cancer cells [5] and the study of their content and functions is relevant mainly for more aggressive tumors such as GBM, providing remarkable insights into intercellular communications during tumor initiation, growth, and recurrence, as well as allowing new possibilities for the diagnosis and treatment of GBM [6,7].

Since GSCs are deeply involved in GBM aggressiveness, recent studies have begun to investigate the compounds specifically transported by EVs and secreted from these cells, which would be implicated in GBM malignancy [8]. We also have recently isolated MVs and EXOs from the culture medium of GSCs obtained from human GBM surgical specimens. Focusing on EV proteome, we identified specific proteins for each EV subtype, such as chaperones or metabolic enzymes in MVs, as well as proteins playing a role in cellular interaction with the extracellular matrix and in the cellular capacity for infiltration and resistance to chemotherapy treatments in EXOs [9]. Here, we investigated the proteomic content of GSCs-derived MVs and EXOs following pharmacological stimulation of the ionotropic P2X7R with low affinity to ATP [10], based on the evidence that: (i) extracellular ATP levels in glioma cells are in the high micromolar range due to the poor nucleotide metabolism [11]; (ii) decreased ATP levels positively correlate with enhanced GBM growth [12]; (iii) P2X7Rs sense high extracellular ATP levels and promote glioma progression [12]; (iv) P2X7Rs are highly expressed in GSCs [13] and their stimulation increases cell aggressiveness by enhancing the expression of epithelial-mesenchymal transition (EMT)-related markers, cell migration/invasion, and GSC survival [14]. Indeed, our results showed that the P2X7R stimulation markedly changed the content of EVs secreted from GSCs, at least in vitro, by mostly increasing the expression of proteins often associated with the progression of tumors and, possibly, their resistance to therapeutic treatments.

## 2. Materials and Methods

### 2.1. Materials and Chemicals

Culture medium, antibiotics, and antifungal drugs (penicillin/streptomycin and amphotericin B, respectively), as well as 2′[3′]-*O*-[4-benzoylbenzoyl]adenosine-5′triphosphatetri[triethylammonium] salt (BzATP) and most chemicals, unless differently specified, were from Sigma-Aldrich S.p.A. (Milan, Italy); tissue culture disposable materials were from Falcon (Corning, Turin, Italy).

### 2.2. EV Isolation from Cultured GSCs by Sequential Centrifugal Ultra-Filtration

EVs have been obtained from the culture medium of GSCs isolated from 2 different patients with primary GBM (identified by an internal numeration as patients #1 and #83), who agreed to be enrolled in the research protocol by signing the informed consent (Prot. 4720/17 approved on 16 March 2017 by the institutional Ethics Committee of the School of Medicine, Catholic “Sacro Cuore” University). The cells, which have already been used in our previous studies [13,14], were characterized for their self-renewal potential, stemness marker expression, and chemotherapy resistance (for details, see [15,16,17]). GSCs were initially grown as nonadherent neurospheres in Dulbecco’s Modified Eagle’s Medium/Nutrient Mixture F-12 Ham (DMEM/F-12) without serum, to which human recombinant epidermal (EGF, 20 ng/mL) and fibroblast (FGF, 10 ng/mL) growth factors (PeproTech, SIAL, Rome, Italy) were added. For the experiments, 2 × 10^9^ cells, obtained from passages 5 to 10 without significant changes in their morphology, were seeded on culture dishes pre-coated with Matrigel (Corning, SIAL), which allowed a uniform pharmacological treatment of GSCs growing as a monolayer, while leaving the cell spherogenic properties unaltered [15,16,17]. GSCs were fed for 48 h with the culture medium mentioned above, which was replaced for the next 48 h with DMEM/F-12 Ham medium without phenol red (Sigma-Aldrich, Milan, Italy), supplemented with the same mitogens. From 48 h up to 96 h, cells were exposed to BzATP (100 μM) or untreated (control). At 96 h, the culture medium was removed and used for EV isolation. For this aim, we used the published procedure [9]. Briefly, 30 mL of medium from each dish were centrifuged at 600× *g* for 10 min and at 4 °C. The supernatant (that is, cell secretome) was then concentrated to 2 mL and filtered (Amicon Ultracel3K Millipore, Merck KGaA, Darmstadt, Germany). Subsequently, part of this concentrated/filtered secretome was centrifuged at 100,000× *g* after washing with phosphate buffered saline (PBS). The pellet obtained was then characterized as an Fn1 fraction, containing MVs. An equivalent part of the concentrated medium was submitted to sequential filtration using pore-sized ultrafilters from 0.65 to 0.45, 0, 22, and 0.1 µm (Durapore Ultrafree CL, Merck Millipore, Merck Life Science, Milan, Italy), giving rise to a final fraction called Fn5, then characterized as containing EXOs. The pellets obtained from the first (Fn1) and fifth (Fn5) fractions were used to perform the subsequent analyses. The sample protein concentration was assayed by the Pierce method [18].

### 2.3. Two-Dimensional Gel Electrophoresis (2DE) Analysis

To analyze the proteome changes in MVs and EXOs isolated from GSCs submitted to the P2X7R pharmacological stimulation, all 2DE experiments were carried out on biological and technical replicates. Totals of 150 μg (for the analytical gels) and 500 μg (for preparative gels) of proteins obtained from MVs (Fn1) and EXOs (Fn5) were mixed with rehydration solution (GE Healthcare, Uppsala, Sweden). Analytical gels were stained with ammoniacal silver nitrate, while gels for MALDI-TOF/TOF mass spectrometry (MS) were glutaraldehyde-free silver-stained, as previously described [19]. Gels were scanned and digitized by GS_900 Calibrated Densitometer (BIO-RAD laboratories, Hercules, CA, USA). The resulting images were analyzed by 2D Platinum Image Master software (version 6.0, GE Healthcare Life Sciences, Uppsala, Sweden).

After background subtraction, the intensity volume of each spot was normalized with respect to the intensity volumes obtained from all spots within the same 2-D gel and was then matched across the different gels. The resulting data were expressed as the mean ± SEM and analyzed by multiple comparisons using one-way analysis of variance (ANOVA). Even though a probability (*p*) value < 0.05 was considered statistically significant, only protein spots with *p* value < 0.001 were selected for identification by MALDI-TOF/TOF MS.

### 2.4. Mass Spectrometry Analysis

Protein spots, selected as indicated above, were excised from gels and analyzed by using a peptide mass finger printing (PMF) approach with a MALDI-TOF/TOF spectrometer. Each picked spot was washed with ethanol (100%) and ammonium bicarbonate (NH_4_HCO_3_, 100 mM), and then incubated in 100 µL of 50 mM NH_4_HCO_3_ supplemented with 10 mM DTT (60 min, at 56 °C), followed by 30 min at room temperature and in the dark in 100 µL of 50 mM NH_4_HCO_3_ plus iodoacetamide. Finally, the gel was reswollen in 50 mM NH_4_HCO_3_ containing porcine trypsin (Promega Italia, Milan, Italy) and incubated at 37 °C overnight. The peptides so extracted were concentrated and desalinated by chromatography (C18ZipTip microsystem, Millipore, Bedford, MA, USA) using 0.1% trifluoroacetic acid (TFA) for repeated washing and eluted in 0.5 μL of a solution of α-cyan-4hydroxycinamic acid (HCCA) and 0.1% TFA (1:1). The obtained eluate was applied on ground-steels and submitted to MS analysis (AUTOFLEX Speed MALDI-TOF/TOF MS instrument, Bruker Daltonics, Brema, Germany). MS apparatus was calibrated by standard molecules including bradykinin (fragment 1–7, 757.39 *m*/*z*), angiotensin II (1046.54 *m*/*z*), ACTH (fragment 18–39, 2465.19 *m*/*z*), [Glu-1]-fibronepeptide B (1571.57 *m*/*z*), and porcine renin tetradecapeptide substrate (1760.02 *m*/*z*). The investigated proteins produced a spectrum in PMF analysis with a range beyond *m*/*z* 700–3000 Da. The internal mass calibration was performed by the trypsin autolysis products (842.50 *m*/*z*, 1045.56 *m*/*z*, 2211.11 *m*/*z*, 2283.19 *m*/*z*). Contaminant trypsin and keratin peaks were removed from the peak list. Data obtained from PMF analysis were entered into known databases (NCBI and Swiss Prot) and analyzed by the Mascot search engine, to compare the masses obtained from the tryptic digest with the theoretical masses found in the databases. Various research parameters were used for MS analysis including PMF, trypsin, fixed modifications (carbamido-methylation), variable changes (i.e., methionine oxidation), monoisotopic mass, state of charge of the peptide +1, the number of maximum errors in the peptide cutting up to 1, and mass tolerance for each peptide (100 ppm, 0.6–0.8 daltons).

Protein assignment was validated by LIFT-MS/MS technology, by which the most abundant proteins were selected and analyzed as ions, choosing a number of precursor ions per sample equal to four. PMF and MS/MS data were then combined in the BioTools 3.2 program connected to the Mascot search engine. The protein identification was considered as univocal when the match between experimental data and sequences deposited in the database showed a *p* value < 0.05 (probability score) (see Appendix A). The scores were reported as log10 (*p*), where *p* represents the maximum probability. The acceptable score value was set at 70 for PMF and 30/40 for MS/MS research.

### 2.5. Bioinformatic Analysis

Data obtained by the MS technology were also analyzed through further databases such as Protein Analysis Through Evolutionary Relationship (PANTHER), Gene Ontology (GO), and UniProt, to determine some characteristics of the EV proteins with expression modified by GSC exposure to P2X7R stimulation, i.e., their molecular function and pathways in which they were involved. The same proteins were also imported into the software STRING (http://string-db.org/, accessed on 15 February 2024) to highlight protein–protein interactions as well as to better define the functions and pathways associated with them. The analysis cut-off was equal to a confidence level of 95%.

### 2.6. Data Analysis

Numerical values reported in the Section 3 (Results) are expressed as the mean ± S.D. (standard deviation). Data obtained from at least two independent biological replicates were analyzed for statistical significance, considering differences as statistically significant at a *p* value < 0.05 (*t* Student, one way).

## 3. Results

The experiments were performed on GSCs derived from the primary GBMs of two patients, already used in previous papers [9,13,14]. Cultured cells were either untreated (control) or exposed to the treatment with a rather selective P2X7R agonist, BzATP, at 100 µM for 48 h. This experimental protocol did not affect cell viability while increasing GSC aggressiveness [14]. Using the ultrafiltration technique (see Section 2), we have then isolated two fractions from the GSC-derived culture medium, which we have previously characterized as MVs and EXOs by their morphological ultrastructure and protein markers using electron microscopy and western blot analysis, respectively [9]. These particles were not contaminated by plasma components, as GSCs normally grow in the absence of serum, or by cell debris, which were removed by the first centrifugation performed during the EV isolation procedure.

### 3.1. Bidimensional (2D) Electrophoretic Analysis of the Protein Cargo of MVs and EXOs Isolated from the Culture Medium of Control- and BzATP-Treated GSCs

First, we measured the total protein amount present in the EVs obtained from each of the two GSC types, both grown either in the control condition or under BzATP treatment. We found a protein quantity of approximately 1.33 ± 0.008 and 0.714 ± 0.004 mg for control GSC-derived MV and EXO fractions, respectively, and 1.75 ± 0.012 and 1.95 ± 0.004 mg for the same fractions derived from BzATP-treated GSCs (n. of samples assayed for each experimental condition and GSC type = 3). Subsequently, 150 μg of total proteins for each fraction were run by 2D electrophoresis on 12% gel (4–7 pH gradient) that resolved 1870 ± 25 and 1733 ± 36 protein spots for the control GSCs-derived MV and EXO samples, respectively, and 2003 ± 115 and 1799 ± 81 protein spots for MV and EXO fractions from Bz-ATP-treated GSCs. Bidimensional maps representative of those obtained from MV and EXO proteins are reported in Figure 1A,B.

The gel image analysis showed a similarity greater than 92% in the expression of the proteins within the vesicular fractions derived from the same GSC sets, that is control or BzATP-treated cells. This similarity resulted by the similar number of protein spots and matching percentage (%) between gels from each fraction of the same experimental set.

This analysis confirmed the exclusive expression of 471 ± 10 and 2675 ± 19 spots in the MV and EXOs, respectively, from control GSCs, as previously observed [14]. However, there was also a significant similarity between MV and EXO proteomes from control (89% similarity due to common spots). These proteins can likely be considered as constitutive cell proteins.

Of all protein spots identified in gels from control MVs or EXOs, only those showing an expression level ≥ 2 (called TOP proteins) coupled to an intensity value statistically significant (*p* < 0.05) were selected for the comparison with proteins from MV and EXO gels from BzATP-treated GSCs. Although many proteins listed in Table 1, Table 2 and Table 3 were identified by this analysis, only some of them have been reported in Figure 2 as an example, taking into consideration three spots for each 2D map related to control and P2X7R-stimulated MV or EXO proteins. Changes in expression level were quantified by comparing the protein intensity of the corresponding spots (Figure 2). Such a comparative analysis revealed significant differences. In particular, we observed the presence of some proteins/spots only in each of the two EV fractions from BzATP-treated cells (reported as ex novo-induced proteins in Table 1 and Table 3, upper part). The GSC stimulation of P2X7Rs also caused a modification in the expression of some TOP proteins selected in control, either as distinctive proteins for each fraction (Table 1 and Table 3, lower part) or as proteins in common between MVs and EXOs (Table 2), mostly upregulating them. Based on this evidence, a restricted number of protein spots, selected based on their expression level (*p* < 0.001), were picked from MV and EXO gels and submitted to digestion by trypsin followed by identification in MALDI-TOF-TOF mass spectrometry (MS).

### 3.2. Influence of the P2X7R Stimulation in Cultured GSCs on the Expression of TOP Proteins in MVs

As stated above, P2X7R stimulation of GSCs induced the ex-novo expression of some MV proteins, reported in the upper part of Table 1, while the expression of other proteins, which was generally low in MVs isolated from the secretome of untreated cells, was mostly upregulated, except that of cytochrome b-c1 complex subunit 1 (QCR1), serine/threonine-protein phosphatase 2A 65 kDa regulatory subunit A alpha isoform (2AAA), an isoform (23) of VIME and triosephosphate isomerase OS (TPIS), which resulted in downregulation as compared to control. However, other VIME isoforms in MVs (47) as well as in EXOs (namely isoforms 6 and 65, see Table 3), were remarkably ex novo induced and enhanced, respectively, by P2X7R cell stimulation.

MS analysis also revealed that some proteins in MVs from GSCs exposed to BzATP were detected more than one time in gel spots, such as dihydropyrimidinase-related protein 2 (DPYL2), copine-1 (CPNE1), or vimentin (VIME), which are isoforms of the same protein with a different PI and, likely, a different function (Table 1). The same is valid as for peroxiredoxin-4 (PRDX4), which is also present among the proteins in common between MVs and EXOs (Table 2).

The intracellular origins of the proteins reported in Table 1 are variable, like that of all proteins in Table 2 and Table 3, deriving for example from nuclei (laminin B1, LMNB1), mitochondria (stress-70 protein, GRP75, also known as mortalin, localized also in nucleus, cytoplasm and endoplasmic reticulum, and ATP synthase subunit beta, ATPB), or cytoskeleton (VIME, DPYL2 also known as collapsin response mediator protein-2, CRMP2). LMNB1 is involved in nuclear functions including DNA replication, transcription, and repair [20], while GPR75 mainly regulates mitochondrial biogenesis and also contributes to maintaining cellular homeostasis (reviewed in [21]). DPYL2 modulates neuronal development, polarity, and growth, and is involved in cell migration. It is also necessary for cytoskeleton remodeling together with VIME, which regulates the mechanical properties of cells and also signal transduction, motility, and inflammatory responses [22].

Other proteins ex-novo induced in MVs by GSC treatment with BzATP are membrane proteins, such as annexin 2 (ANXA 2) and annexin 5 (ANXA5), while the heath shock protein 60 (CH60) and ATP synthase subunit beta (ATPB) are mainly located in mitochondria. ANXA2 is a calcium-regulated membrane-binding protein, which modulates cross-linking of plasma membrane phospholipids with actin and the cytoskeleton, also regulating exocytosis, while ANXA5, which plays a physiological role in the blood coagulation cascade as an inhibitor of thromboplastin activity, can also promote cell membrane repair [23]. CH60 is a chaperonin mainly acting at mitochondrial level for protein import and functional arrangement into macromolecules. However, CH60 can also be localized at the plasma membrane level of mammalian cells, including human tumor cells, where it modulates various activities ranging from phagocytosis to cell adhesion and migration through the extracellular matrix [24]. ATPB derives from mitochondrial membranes contributing to ATP synthesis by interacting with the subunit alpha of the catalytic core of the F1 domain of ATP synthase. Of note, another mitochondrial protein, the protease ATP homolog (ATP23), usually located in the mitochondrial inner membrane (Table 2), was detected in both EV fractions. ATP23 is involved in double-strand break repair and can act as a metalloprotease crucial for mitochondrial ATPase biosynthesis [25].

In the lower part of Table 1, we reported the proteins whose expression was modified by GSC exposure to BzATP. Among those with upregulated expression, there are peroxiredoxin 4 (PRDX4) and peroxiredoxin 2 (PRDX2), which, however, were detected in both MVs and EXOs (Table 1 and Table 2). They belong to the family of PRDX, comprising six enzymes with peroxidase activity aimed at maintaining intracellular reactive oxygen species (ROS) homeostasis [26].

Other upregulated proteins in Table 1 are more related to cytoskeleton organization and cell motility, such as actin cytoplasmic 1, also known as actin β (ACTB), F-actin-capping subunit beta (CAPZB), and F-actin-capping subunit alpha 1 (CAZA1). ACTB is one of the six actin isoforms. In particular, ACTB and actin γ are two non-muscle cytoskeletal actins [27], exerting different functions: β-actin is involved in cell contractility, while γ-actin participates in cell motility [28]. Of interest, we observed the overexpression of ARP3, a protein in common between MVs and EXOs (Table 2), which regulates actin filament polymerization together with ARP2, both belonging to the ARP2/3 complex [29].

Proteasomal proteins were also upregulated by GSC stimulation of P2X7Rs, including the proteasome activator complex subunit 1 (PSME1) (Table 1), which is expressed in the cytoplasm and nucleus and is involved in immunoproteasome assembly that is crucial for correct antigen processing [30]. The proteasome subunit alpha type 6 (PSA6) and the COP9 signalosome complex subunit 4 (CSN4) are listed as common proteins between MVs and EXOs (Table 2). PSA6 is a member of the 20S core proteasome complex, which contributes to the proteolytic degradation of most intracellular proteins by the ATP-dependent degradation of ubiquitinated proteins, aimed at maintaining protein homeostasis [31]. As for CSN4, it belongs to the mammalian CSN complex, which regulates cell proliferation and survival, regulating protein degradation via the ubiquitin-proteasome pathways [32].

The remaining upregulated proteins of Table 1 play rather heterogeneous functions. In particular, copine 1 (CPNE1) is a calcium-dependent phospholipid-binding protein present in the cell nucleus, cytoplasm, and plasma membrane [33]. Additionally, RuvB-like 2 (RUVBL2) is usually, but not exclusively, located in cell nuclei and belongs to the RuvB-like family, which shows multiple physiological activities related to chromatin remodeling and transcriptional regulation [34]. Lastly, HS90B is a chaperone, which, by its ATPase activity, controls the functionality of specific target proteins involved in cell cycle and signal transduction [35].

In contrast, only a few proteins were downregulated upon GSC treatment with BzATP. Of these, besides an isoform of VIME, QRC1 is a member of the ubiquinol-cytochrome c oxidoreductase involved in the mitochondrial electron transport chain driving oxidative phosphorylation. 2AAA is the PR65 subunit of the protein phosphatase 2A (PP2A), which modulates cell cycle, development, and growth, regulating numerous molecular pathways as well as cytoskeleton rearrangement and cell mobility. TPIS is an enzyme responsible for the conversion of dihydroxyacetone phosphate (DHAP) into D-glyceraldehyde-3-phosphate (G3P) in glycolysis and gluconeogenesis [36] and also for the production of methylglyoxal (MGO), a cytotoxic side-product of those reactions able to alter proteins, DNA, and lipids.

Finally, GSC exposure to BzATP modified the expression of two additional proteins in common between the two EV fractions (Table 2). One of these, ferritin light chain (FRIL) was upregulated. It is one of the two subunits forming unbound ferritin, which is linked to iron homeostasis and delivery to cells. Ferritin is also known to influence tumor microenvironment and immunity and is involved in ferroptosis, a type of programmed cell death [37]. In contrast, the other protein, chloride intracellular channel protein 1 (CLC1), was downregulated by P2X7R stimulation of GSCs. It belongs to the p64 family of chloride channels, which are present in cells also as cytosolic molecules able to interact with cytoskeletal proteins. Of note, CLIC1, when it is in the soluble form, shows an enzyme activity similar to glutathione S-transferase [38].

### 3.3. Influence of P2X7R Stimulation of GSCs on the Expression of TOP Proteins in EXOs

Even in Table 3, VIME was detected in gel spots more than one time as isoforms of the same protein. Furthermore, the expression of the first seven proteins in Table 3 (upper part) was ex novo induced in EXOs, while the expression of the other proteins (lower part of Table 3) was mostly upregulated by cell treatment with BzATP, with the exception of ATPB, an isoform of VIME (63), and ACTB, which was downregulated as compared to control.

Like the proteins listed in Table 1 and Table 2, most of those included in Table 3 are involved in physiological cell processes as well as in tumor development and/or growth. Since the normal function of some of them has been reported in the previous paragraph, here we focused on proteins not yet examined.

Among the proteins ex novo induced in EXOs by GSC treatment with BzATP, there are four proteasomal proteins belonging to the 20S core proteasome complex and deputed to the ATP-dependent metabolism of ubiquitinated proteins such as the proteasome subunits alpha 2, 5, and 6 (PSA2,5,6) and the proteasome subunit beta 9 (PSB9), of which PSA6 is also reported in Table 2 [39].

Other BzATP-induced proteins are elongation factor 1 gamma (EF1G) and histone binding protein RBBP4. EF1G is one of the three proteins forming the elongation factor complex, which assures the delivery of aminoacyl tRNAs to the ribosomes [40], while RBBP4 (also known as RbAp48, or NURF55) is a nuclear protein implicated in histone acetylation and chromatin assembly [41].

In the lower part of Table 3, besides DPYL2, ATPB, VIME, and ACTB, already examined, two further proteins are linked to cell cytoskeleton and are actin cytoplasmic 2 (ACTG) and Tropomyosin alpha 4-chain (TPM4). The former, also known as actin γ, modulates together with ACTB cell cycle and proliferation [42], while the latter binds to actin filaments to stabilize them in the cytoskeleton [43].

Some other proteins are instead related to extracellular matrix turnover. One of these is the 72 kDa type IV collagenase, also known as matrix metalloproteinase-2 (MMP-2) or gelatinase A, a zinc-dependent enzyme that, when activated on the cell membrane, can degrade type IV collagen, the most abundant constituent of the basement membrane [44].

In contrast, procollagen C-proteinase enhancer-1 (POC1, often reported as PCPE-1) accelerates procollagen maturation and fibril formation, without hindering the activities of other extracellular metalloproteinases [45]. Another upregulated protein by cell treatment with BzATP was the metallopeptidase inhibitor 2 (TIMP2), which together with the others encoded by the same gene family is a natural inhibitor of the matrix metalloproteinases, a group of enzymes involved in extracellular matrix degradation [46]. Additionally, TIMP2 has the ability to directly suppress the proliferation of endothelial cells, thus assuring tissue homeostasis in response to angiogenic factors.

The other proteins in Table 3 play more heterogeneous physiological roles, mainly in the central nervous system (CNS), apart from the eukaryotic initiation factor 4A1 (IF4A1), which is a subunit of the IF4F complex with a wide role being necessary for mRNA binding to ribosome [47]. Thus, MPP2 membrane protein, palmitoylated 2 (MAGUK p55 subfamily member 2), is a structural constituent of postsynaptic density membrane, involved in excitatory postsynaptic potential and long-term synaptic potentiation. It is located in the dendrite and glutamatergic synapse [48]. Protein disulfide-isomerase A3 (PDIA3), also known as glucose-regulated protein, 58-kD (GRP58), is a ubiquitous isomerase enzyme of the endoplasmic reticulum, where it cooperates with calreticulin and calnexin for a correct folding of neosynthesized glycoproteins [49]. Accordingly, its dysregulation has been reported in several neurodegenerative diseases (i.e., [49,50]). Gamma-enolase (ENOG), also indicated as specific neuronal enolase (NSE), is one of three isoenzymes in the glycolytic pathway identified in mature neurons and in cells of neuronal origin [51]. Additionally, serpin B6 (SPB6, also known as serine proteinase inhibitor B6) may be involved in the regulation of serine proteinases present in the brain. SPB6 may act as an inhibitor of autophagy, thus contributing to maintaining tumor cell growth in metabolic stressful conditions [52]. Finally, carboxypeptidase E (CBPE) was initially reported as an enzyme involved in prohormone processing [53]; however, it was subsequently shown that CBPE is provided with a non-enzymatic activity acting as a trophic factor for neuronal survival [54].

### 3.4. Validation of Protein Sequence Identification by MS/MS

We performed an additional analysis to validate the identification of the proteins reported in Table 1, Table 2 and Table 3. We applied LIFT technology (reported in the Section 2 and Appendix A) to our protein samples, through which it was possible to obtain ion parental masses from PMF spectra. This method allowed us to identify peptide sequences (reported in Table 4, last column on right) specific for each protein selected from the cargo of MVs and EXOs derived from control and P2X7R-stimulated GSCs. Given the huge number of sequences determined for all proteins under investigation, here we reported some examples related to the validation of the proteins reported in Figure 2. Looking at Table 4, the high values obtained for Score Tof-Tof ensures the uniqueness of the protein sequence and, thereby, its identification.

### 3.5. Influence of GSC Exposure to the Stimulation of P2X7Rs on the Functional and Biological Activities of the Proteins Isolated from EXO and MV Fractions

The proteins with increased expression in MVs and EXOs induced by GSC exposure to P2X7R stimulation were analyzed by using the GO and PANTHER databases to highlight and summarize the biological processes and molecular pathways in which they are usually involved (Figure 3A,B).

The comparison of the percentage distribution of the vesicular proteins mentioned above among biological processes (Figure 3A) highlighted major differences in the role played by MVs and EXOs derived from P2X7R-stimulated GSCs. Indeed, most MV proteins were deputed to cellular processes related to cytoskeleton or nuclear organization and their biological regulation, while a small percentage of them played a role in metabolic processes and the response to noxious stimuli such as oxidative or inflammatory stress. In contrast, EXO proteins were mostly linked to cellular processes related to protein turnover, while a lower percentage of them showed an involvement in nuclear processes in response to stimuli or in biological regulation of extracellular matrix. In both EV fractions, however, there was a similar percentage of proteins with a not well-defined heterogeneous role (11–12%).

Again, the PANTHER GO analysis showed the relationship of the proteins with enhanced expression caused by P2X7R stimulation in GSCs with different molecular pathways in the two EV fractions (Figure 3B). Thus, in both EV fractions, there was an almost similar and higher percentage of proteins related to cytoskeletal regulation by Rho GTPase, which has widely been associated with human cancer [55,56]. The other proteins were mainly related, although each at a small percentage, to ATP synthesis and glycolysis, inflammatory pathways, some neurodegenerative diseases, and also signals, including integrin, cadherin, FAS, FGF, and Wnt pathways, which are, in turn, coupled to cancer enhancement and progression [57,58,59]. Lastly, in EXOs, a small percentage of proteins was also linked to G protein activated-pathways.

The details on the differences indicated above are shown in Appendix A.

Overall, this analysis emphasizes the complementarity of the vesicular proteins with P2X7R-enhanced expression in promoting cellular cytoskeletal rearrangement, modulating cellular metabolism and extracellular matrix degradation, and enhancing protection against stressful stimuli. Therefore, if transferred to sister cells, they could contribute to the spread of GBM.

## 4. Discussion

It is increasingly evident that the purinergic ionotropic P2X7Rs are intimately implicated in the growth and metastases of many tumors [60]. P2X7Rs appear to be also involved in the release of EVs from immune and cancer cells, even though this aspect is still incompletely investigated (reviewed in [61]). Our previous and present data support the oncogenic role of P2X7Rs, as their stimulation could favor GBM malignancy. Indeed, we have demonstrated that: (i) GSCs, a distinct subpopulation of tumor cells playing a crucial role in GBM aggressiveness/recurrence [62], show high expression levels of P2X7Rs [13]; and (ii) P2X7R stimulation of these cultured cells leads to upregulated expression of classic EMT markers, increased cell migration, and invasiveness in vitro, without affecting GSC viability [14]. In parallel, it has been reported that GBM releases EVs, which act as carriers of oncogenic proteins to be transferred to surrounding tumoral cells, favoring their survival and expansion [63]. In agreement with these data, we have identified two EV subtypes, MVs and EXOs released from GSCs, and characterized the most expressed proteins in their cargo as involved in metabolic support and increased aggressiveness in tumor cells [9]. Here, we have shown that the P2X7R stimulation of GSCs could mostly stimulate the expression of several proteins in MVs or EXOs as compared to those identified in both EV fractions from control GSC secretome. If this finding obviously accounts for the increase in the total protein amount measured in the same EV fractions, it is of note that most of the ex novo induced/upregulated proteins have already been recognized as tumoral biomarkers and/or tumorigenic factors in GBM, while for those not yet recognized as potential GBM cancerogenic targets, further research needs to be done. In this regard, it should be emphasized that: (i) our findings have been obtained using human GSCs derived from patients with GBM, while most papers on the same topic report results obtained on immortalized cell lines (i.e., [64,65]); (ii) in tumors there may be hundreds of gene mutations, many of which could be potential targets for discovering novel biomarker candidates (reviewed in [66]).

The EV proteins with enhanced expression caused by GCS stimulation of P2X7Rs were closely related each other in an intense functional network as shown in Figure 4 and Figure 5, obtained using the STRING software. This analysis also indicated that they are interconnected with many other proteins. In particular, MV proteins in Table 1 are likely related with chaperonins, involved in protein folding, or with enzymes provided with phosphatase activity, while a few others are implicated in the control of the cell cycle or mitochondrial activity (see Appendix A). In contrast, EXO proteins reported in Table 3 are closely related to proteasomal proteins (Appendix A).

These networks can reinforce the oncogenic role played by the EV proteins with enhanced expression, if transferred to tumor cells. Many of them are related to cytoskeleton reorganization and cell migration, including LMNB1, VIME, DPYL2, ACTB, ANXA 2 and 5, CAPZB, and CAZA1 found in MVs, as well as ARP3 identified among proteins in common between MVs and EXOs and TPM4 together with ACTG in EXOs. About their oncogenic potential, an elevated expression of LMNB1 and related proteins of the same nuclear lamina family has been associated with tumor development, aggressiveness, and metastasis in a wide variety of cancers [67], as well as in glioma cells [68]. Furthermore, gliomas generally possess increased amounts of the intermediate filaments of VIME [69], which may have a role in migration of glioma cells, especially following their exposure to radiation [70]. DPYL2, which is a promoter of microtubule assembly, shows increased expression levels in GBM, being recognized as a “glioma reference biomarker” together with ACTB and other proteins [71], the overexpression of which is currently under intense investigation for their involvement in tumor cell migration and metastasis, mainly in glioma [72]. Connected to the cytoskeletal function of actin are ARP3 and TPM4. ARP2/3 expression has been related to an oncogenic role in colorectal, adenocarcinomas, gastric tumors (reviewed in [73]), and also in GBM [74]. A recent proteomic analysis of GBM-derived EVs has identified ARP3 among the proteins involved in filopodia formation, mainly in EXOs deriving from the most aggressive GBM types [64]. Interestingly, ARP2/3 might also be involved in the maintenance of the glioma cell stemness [75]. Additionally, TPM4, the suppression of which can inhibit metastasis in in hepatocellular carcinoma (HCC) [76], showed significantly higher levels in glioma than in healthy brain tissue, which correlated with poor prognosis of patients [77]. Likewise, ANXA2, which has been isolated in EVs from glioma cell lines and patients, is an important mediator of EV-cell interactions able to promote tumor angiogenesis [78] and the circADAMTS6/ANXA2/NF-κB axis plays an important role in accelerating GBM growth in a hypoxic microenvironment [79]. Differently, ANXA5 is regarded as a potential early biomarker in hepatocarcinogenesis together with ANXA2 [80] and a predictive biomarker for tumor progression in different cancer types [81], but it was not so far identified as a specific oncogenic protein in gliomas as well as CAPZB and CAPZA1. CAPZB overexpression has indeed been reported in epithelioid sarcoma tissue specimens, where it increased cell growth and motility [82], while CAPZA1 could regulate cell growth, invasion, and EMT markers in various human tumors, representing an unfavorable prognostic marker in gastric and lung cancers [83,84,85].

All these findings, besides reinforcing our previous results on the increased aggressiveness of GSCs exposed to P2X7R stimulation, would attribute a key role to EVs derived from BzATP-treated GSCs in cytoskeleton reorganization involved in the GBM metastasis process. Indeed, cytoskeletal proteins are fundamental in tumor cell spreading, which is largely dependent on cytoskeleton reshaping as for infiltration of glioma cells [86].

P2X7R stimulation of GSCs also increased the EV expression of antioxidant and proteasomal enzymes belonging to the family of PRDXs and 26S proteasome, respectively. The PRDX contribution to tumorigenesis and cancer recurrence, due to antioxidant activity through ROS scavenging [87], has been substantially demonstrated in numerous tumors [88,89,90]. In particular, overexpression of PRDX1, PRDX4, and PRDX6 has been reported in most histological glioma types compared to the normal tissues and correlated with poor survival of glioma patients [91]. As for the proteasomal enzymes, proteomic data indicated that PSME1 is a potential tumor marker in different human tumors [92,93,94]. Likewise, all the proteasome alpha subunits, including those found in EXOs (which, in turn, are connected with many other proteasomal proteins, as highlighted in Figure 5 and Appendix A), are upregulated in many tumors [95], while PSB9 hyperexpression has been correlated with poorer prognosis in low grade glioma [96].

In agreement with these findings, growing evidence shows that ubiquitination or de-ubiquitination of proteins performed by proteasomal enzymes are important processes of post-translational modifications that, by modulating the turnover of metabolic enzymes and molecular signals, can enhance tumor progression [97].

Additionally, there is evidence that mitochondria are involved in cancer cell metastatic processes [98] and mitochondrial components, including proteins, are released within EVs, even though there are differences in the EV mitochondrial content [99]. Interestingly, radiation promotes the formation of EVs containing mitochondrial proteins from the PC3 (prostate cancer 3) cell line, which could likely enhance survival of these cells after their exposure to radiation [100]. In relation to our findings, we can distinguish two types of mitochondrial proteins with expression enhanced by GSC exposure to BzATP, namely chaperones and ATP-related enzymes. As for the former, CH60 was abundant in brain tumors, although more findings are necessary to identify it as a selective biomarker [101]. Linked to the CH60 activity is that of GRP75/mortalin, a chaperone protein crucial to assist protein folding in mitochondria. Growing evidence indicates that its overexpression may promote cancer cell metastatic invasion and the GRP75–CH60 axis could be a novel target to suppress tumor progression (reviewed in [102]). As for the proteins related to ATP turnover, we found conflicting results. Indeed, P2X7R stimulation of GSCs increased the levels of ATPB, at least in MVs, and ATP23 in both EV types, enzymes which should mutually modulate their activity of promoting and decreasing ATP production, respectively. Of note, ATPB may also exist on the external surface of the cell membrane as ectopic ATP synthase, where it has been indicated as a target marker for tumor therapy [103], while ATP23 was observed in a TCGA (The Cancer Genome Atlas) GBM cohort, in which patients showed a significant decrease in overall survival (OS) [104]. In this context, it is known that cancer stem cells usually show a rapid increase in the aerobic glycolysis when they are in the proliferative phase and the ATP demand is enhanced as well (reviewed in [105]). Thus, these enzymes transported by EVs could serve to implement the metabolism of carbohydrates in GSCs, when received by them. In contrast, some mitochondrial proteins linked to cell energy supply via glucose metabolism were downregulated by GSC exposure to P2X7Rs such as TPIS and QCR1. However, TPIS downregulation could be functional in tumors, since TPIS activity could be responsible for the production of toxic levels of MGO deriving from glucose metabolism [106], while P2X7R-induced QCR1 decreased expression could be compatible with the known mitochondrial alterations reported in GBM, which compromise the energy supply through oxidative phosphorylation [107].

A few other proteins not related to mitochondria were downregulated in EVs from BzATP-treated GSCs, namely 2AAA and CLC1. While, in agreement with our finding, 2AAA, a subunit of PP2A acting as a tumor-suppressor, is inactivated or dysregulated in about 60% of GBM patients [108], CLIC1 is overexpressed in several solid tumors including gliomas (reviewed [109]). The last four proteins were upregulated in MVs from BzATP-treated GSCs. One of these is HS90B, which was also present in EXOs isolated from five GBM cell lines [110], and shows a potential functional network with other important molecules with co-chaperone activities (see Figure 3 and Appendix A). Indeed, HS90B has a role in cancer progression, together with proteins including GRP75 and VIME (reported in our Table 1), and as a potential biomarker, useful for diagnostic and therapeutic applications in malignant melanoma cancer [111] or early grade breast cancer [112]. Furthermore, it has been related to tumors’ chemo-and radio-resistance, also showing angiogenic properties [101]. FRIL, which is overexpressed in high grade glioma, mainly associated with IDH1/2 wildtype and unfavorable prognosis of these glioma patients [113], has also been indicated as a biomarker together with ferritin heavy chain to predict prognosis and temozolomide resistance in glioma patients [114]. The other two upregulated proteins are CPNE1 (with three isoforms), which usually modulates the neural stem cell proliferation and neurite outgrowth, and RUVBL2, involved in chromatin remodeling. They play heterogeneous oncogenic roles, but only in peripheral tumors, [115,116,117] so far.

As for proteins selectively upregulated or induced in EXOs by the P2X7R stimulation in GSCs, many findings have been reported above for proteasomal and cytoskeletal proteins. Looking at the other upregulated EXO proteins, only one, RBBP4, is linked to nuclear activity while EF1G, IF4A1, PDIA3, POC1, CBPE, MMP2, and TIMP2 are mostly related to protein turnover. In this regard, it has recently been reported that protein metabolism is an important contributor to metastasis, amino acids being a primary nutrient source for cancer cells, facilitating their survival and tumor spread [118]. Therefore, it is not surprising that the enzymes indicated above have shown a potential oncogenic role when their expression is increased, although some of them have not yet been related to GBM progression. Thus, changes in EF1G levels have been pointed out in metastatic colon cancer and hepatic HepaG2 cells or cervical cancer specimens from patients in treatment with antineoplastic drugs (reviewed in [119]), while inhibition of IF4A1 may partially inhibit hepatocellular carcinoma progression [120]. In contrast, PDIA3, which participates in cancer initiation, progression, and chemosensitivity [121], has recently been proposed as a novel therapeutic target for GBM therapy [122]. Likewise, although a few papers have reported a role of POC in cancer, its expression was recently correlated with tumor prognosis in 11 tumors, including GBMs [123]. An aberrant upregulation of CBPE has also been found in endocrine (pituitary adenomas) as well as non-endocrine tumors including GBMs [124]. Additionally, high levels of RBBP4, which appears to be related to poor prognosis of colon cancer with or without hepatic metastasis [125], could be also predictive for poor survival outcome of low-grade glioma (LGG) patients and, therefore, it could be regarded as a potential LGG biomarker and an immunotherapy target [126]. The gene codifying for this protein was also identified among other genes that can be used for early diagnosis and more specific GBM treatments [127]. Likewise, the gene codifying for MMP2 has been identified as a novel oncogene in colorectal cancer [128], while increased MMP-2 activity is involved in an essential step for the metastatic progression of most cancers [129] and also linked with a poor prognosis in multiple tumors, including glioma [130]. In contrast, members of the TIMP family, mainly TIMP2, have long been recognized as modulators of MMP activities, thus regulating tissue homeostasis, and suppressing both primary tumor growth and metastasis formation. However, only recent data would indicate other functions beyond the simple inhibition of metalloprotease activity. In particular, the cytokine-like role of TIMPs may also contribute to tissue alterations in various chronic diseases, including tumors (reviewed in [131]).

Finally, some upregulated proteins in EXOs appear to be involved in intercellular signal transduction such as MAGUK, SPB6, and ENOG. An oncogenic role of MAGUK has not yet been reported, while the SPB6 gene was recently found to be upregulated in colorectal cancer [132] and ENOG is considered a pro-survival factor in cancer cells [133]. Interestingly, its activities are regulated by the cysteine peptidase cathepsin X, found together with ENOG in GBM tissues, mainly in macrophages and microglia [134]. Accordingly, ENOG knockdown reduces the migration of GBM cells, sensitizing them to radio- and chemotherapies. In addition, GBM patients with elevated ENOG/NSE expression showed shorter survival than patients with low expression [135].

## 5. Conclusions

On the whole, our findings would imply that the P2X7R stimulation of GSCs mostly enhanced the expression of a number of proteins in GSC-released EVs, which could play oncogenic roles. These proteins have been identified by MALDI-TOF MS analysis and their sequence has been validated by the LIFT technology, a combined method recognized as valid in medicine and research [136,137]. They can likely increase the tumorigenic potential of GSC-derived nanoparticles, in which we previously found chaperones and metabolic enzymes as well as proteins mostly involved in cell-matrix adhesion, cell migration/aggressiveness, and chemotherapy resistance in control condition [9]. Of note, the P2X7R stimulation may occur in GBM in vivo, as the extracellular levels of ATP, the only natural agonist for these receptors, are usually higher in the TME of many tumors, including gliomas, than in the normal corresponding tissues [60].

For many of the proteins with enhanced expression by P2X7R stimulation, the involvement in GBM aggressiveness has already been recognized, while for some others, mainly contained in the EXO fraction, possible roles as tumor biomarkers or therapeutic targets have been reported for tissues or cells from different, mostly peripheral, tumors.

Obviously, we are aware that findings from our study need to be confirmed in EVs obtained from a wider number of GSCs deriving from different human GBMs, in order to validate the increase in the expression of the proteins identified in our study as routinely induced by P2X7R stimulation in human GSCs. If that were the case, this would corroborate findings on the oncogenic role played by these receptors, further promoting research on them as druggable tumoral targets. Indeed, the ultimate goal would be the possibility to treat patients, including those with GBM, with ligands of P2X7Rs. In this regard, the studies are largely at preclinical levels. There are many animal models involving a wide number of animal species, on which it has been possible to study the structure of P2X7Rs, which show a number of variants, and their pathophysiological role in many diseases [138]. Some of these models have very recently been used to check the ability of P2X7R ligands, such as P2X7R monoclonal antibodies, to reduce graft-versus-host disease consequent to allogeneic hematopoietic stem cell transplantation in leukemia and lymphoma [139]. However, early clinical trials have so far produced not positive results. Meanwhile, many P2X7R radioligands have been developed to be used as tracers for imaging and inflammatory responses in CNS disorders, including cancer [140]. This gives rise to hope that P2X7Rs could soon be used as tumor markers for possible early diagnosis of human diseases, until new data allows the use of the ligands of these receptors as new therapeutic agents [141].

## Figures and Tables

**Figure 1 cells-13-00571-f001:**
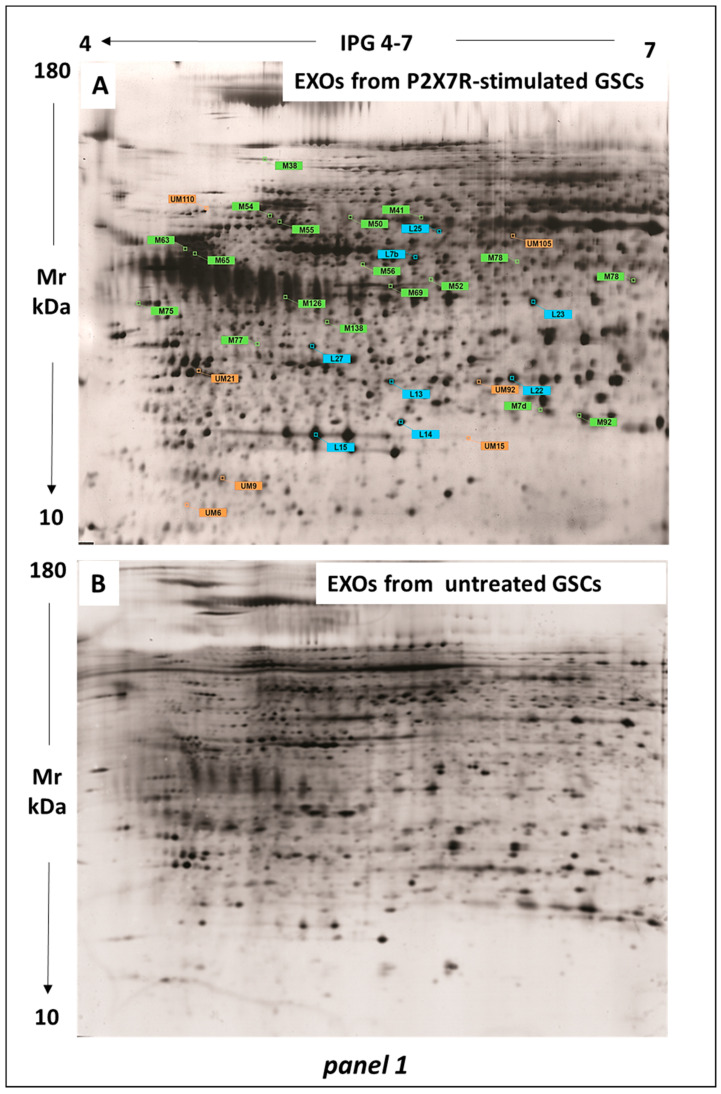
Separation and identification of the major protein spots by bidimensional electrophoresis. Extracts from MV and Exo fractions of GSCs were used, in which the protein expression was modified by the stimulation of P2X7Rs in cultured GSCs as compared with that revealed in the corresponding EVs from control GSCs. (**A**–**D**) electrophoretic profiles of proteins in GSC-derived EXOs (panel 1) and MVs (panel 2). All proteins exclusively expressed in each fraction are marked with: orange labels (UM = unmatched) to indicate those whose expression was ex novo-induced by GSC treatment with BzATP; green labels (M = matched) to indicate those up- or down-regulated by GSC treatment with BzATP; light blue labels (L) to indicate the proteins in common between the two EV fractions with expression modified by GSC pharmacological treatment. UM: unmatched proteins and M: matched proteins in comparison to proteins identified in EXOs from control GSCs. L: indicates the proteins taken as a reference point for the alignment of the other proteins on the image of each gel.

**Figure 2 cells-13-00571-f002:**
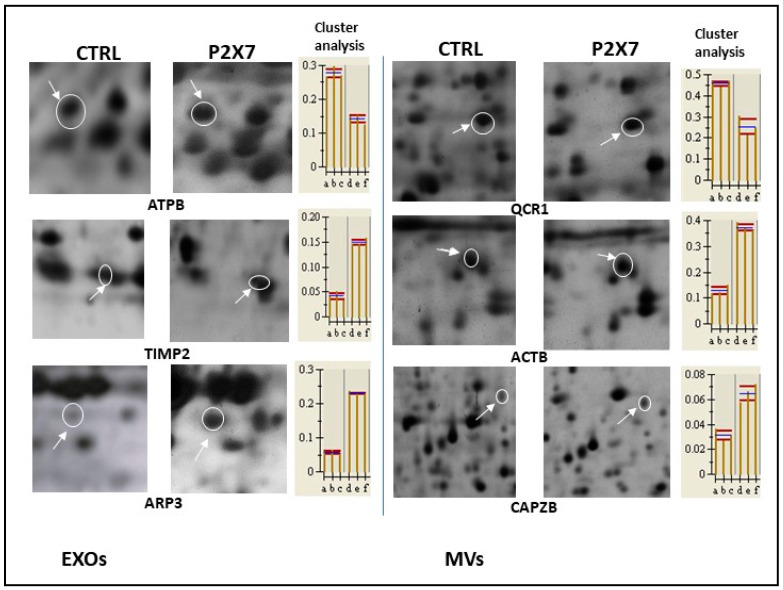
Changes in expression levels of some proteins chosen among those analyzed by MALDI-TOF/TOF MS and reported in Table 1, Table 2 and Table 3. Spots selected from 2D gels, on which proteins from MVs and EXOs obtained from control and P2X7R-stimulated GSCs were run, were compared. In the pictures, these spots, indicated by white circles and arrows, underwent magnification showing spot location within the gels. Their intensity was quantified and the results from heuristic cluster analysis on the related proteins that differ significantly (*p* < 0.001) are reported in the histograms on the right. On the *x*-axis, each letter indicates a single gel (from three biological replicates for each condition) while values on the *y*-axis indicate the relative volume of the spot.

**Figure 3 cells-13-00571-f003:**
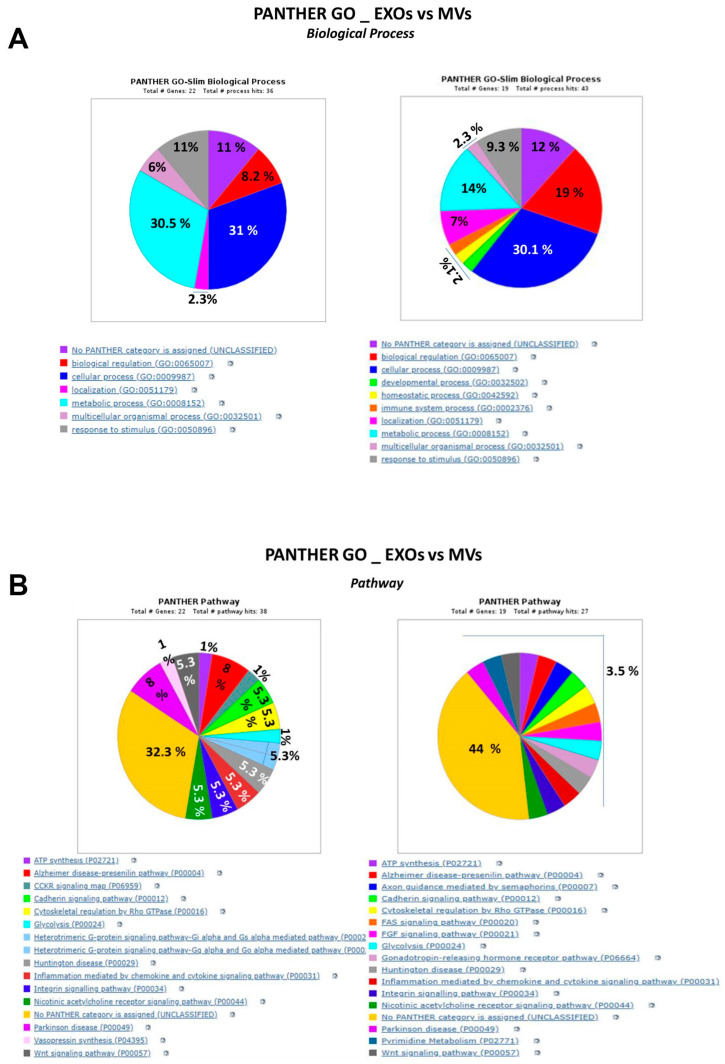
Gene ontology analysis of the proteins identified for each type of EV, which had been ex novo induced or upregulated by P2X7R stimulation of GSCs. This analysis distributed the proteins as percentage based on their involvement in different biological processes (**A**) and pathways (**B**). Pies on left are related to EXO proteins while those on right are related to MV proteins.

**Figure 4 cells-13-00571-f004:**
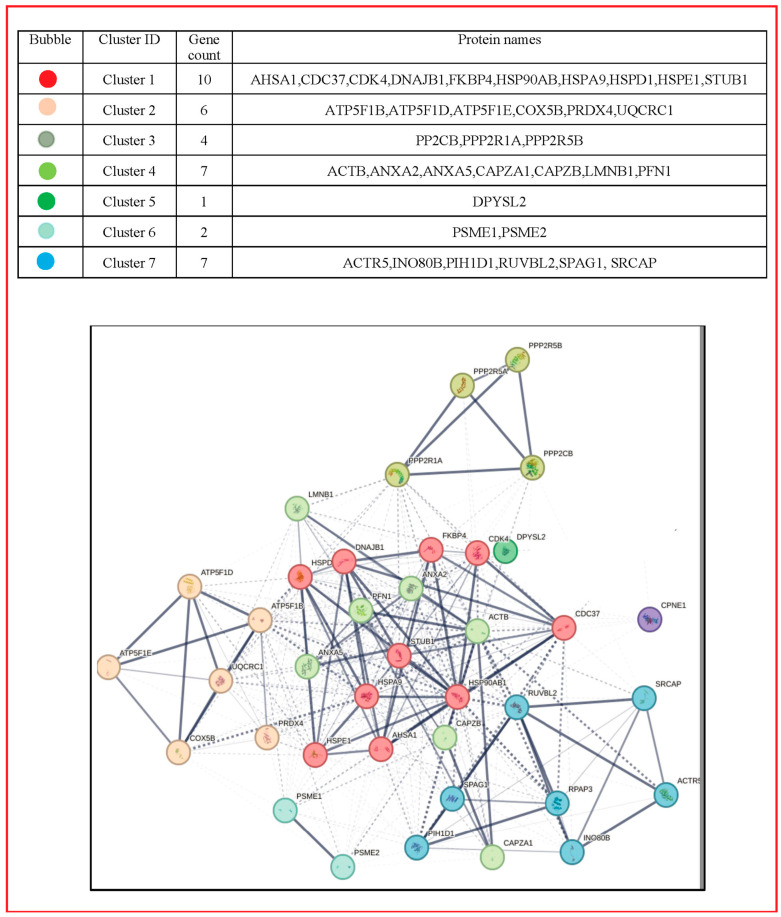
Network of the MV protein interaction. Using STRING (http://string-db.org, accessed on 22 September 2023), the functional links of the proteins expressed in MVs were constructed. Proteins shown as spheres and labeled with gene name represent the nodes, whereas nodes that are associated to each other are linked by edges that represent their interaction. Thicker lines indicate stronger associations. Main clusters are indicated with a colored circle. The color for each circle group is reported in the table above the figure as derived from the STRING analysis. The entire name of the proteins related to the genes in the figure as well as their usual roles are reported in Appendix A, together with the indication of the predicted functional partners and related roles.

**Figure 5 cells-13-00571-f005:**
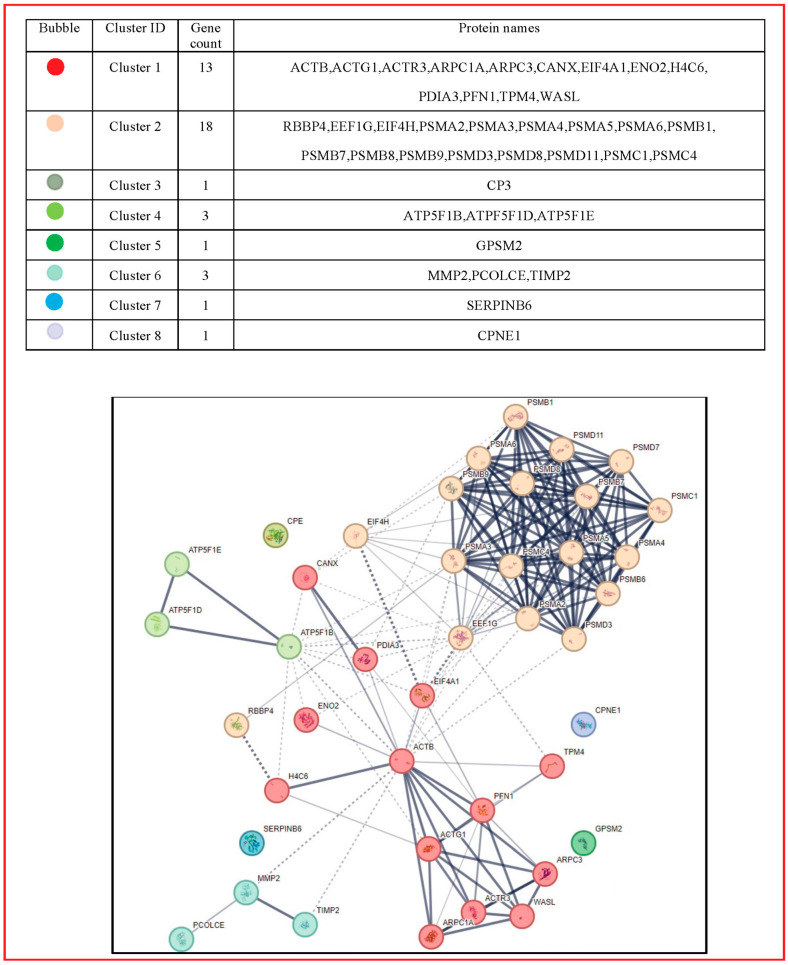
Network of the EXO Protein Interaction. Using STRING (http://string-db.org, accessed on 22 September 2023), the functional links of the proteins expressed in EXOs were constructed. Proteins shown as spheres and labeled with gene name represent the nodes, whereas nodes that are associated to each other are linked by edges that represent their interaction. Thicker lines indicate stronger associations. Main clusters are indicated with a colored circle. The color for each circle group is reported in the table above the figure as derived from the STRING analysis. The entire name of the proteins related to the genes in the figure as well as their usual roles are listed in Appendix A, together with the indication of the predicted functional partners and related roles.

**Table 1 cells-13-00571-t001:** Changes in the expression of proteins in CGS-derived MVs induced by cell stimulation of P2X7Rs.

** *(a) Ex-Novo Induced Proteins by P2X7R Stimulation of GSCs* **										
**SPOT** **ID**	**Abbr.** **Name**	**AC ^a^ Swiss/NCBI**	**Protein Description**	**Score ^b^**	**Peptide** **Matched**	**SC ^c^** **%**	**Theoretical** **(pI/Mr)**	**Experimental** **(pI/Mr)**	***p*-Value**	
UM2	LMNB1	P20700	Laminin B1	87	26	38	5.11–66.65	5.09–69.01	0.0004	
UM10	VIME	P08670	Vimentin	174	58	72	5.06–53.67	5.12–54.04	0.0011	
UM20	ANXA5	P08758	Annexin A5	64	9	29	4.94–35.97	5.00–37.02	0.0008	
UM28	DPYL2	Q16555	Dihydropyrimidinase-related protein 2	85	23	54	5.95–62.71	5.98–63.92	0.0002	
UM31	DPYL2	Q16555	Dihydropyrimidinase-related protein 2	82	23	49	5.95–62.71	6.08–58.75	0.0004	
UM33	CH60	P10809	60 kDa heat shock protein, mitochondrial	91	26	53	5.70–61.18	5.58–60.01	0.0013	
UM34	ATPB	P06576	ATP synthase subunit beta, mitochondrial	125	19	50	5.26–56.52	5.28–54.13	0.0003	
UM37	ANXA2	P07355	Annexin A2	154	32	41	7.57–38.80	7.63–39.06	0.0024	
UM47	VIME	P08670	Vimentin	104	26	45	5.06–53.67	5.34–58.02	0.0008	
** *(b) Changes in Top Protein Levels Caused by P2X7R Stimulation of GSCs* **										
**SPOT** **ID**	**Abbr.** **Name**	**AC ^a^ Swiss/NCBI**	**Protein Description**	**Score ^b^**	**Peptide** **Matched**	**SC ^c^** **%**	**Theoretical** **(pI/Mr)**	**Experimental** **(pI/Mr)**	***p*-Value**	**Variation**
M1	DPYL2	Q16555	Dihydropyrimidinase-related protein 2	112	25	58	7.26–49.85	7.42–49.33	0.0037	UP
M1bis	HS90B	P08238	Heat shock protein HSP 90-beta	140	36	45	4.97–83.55	4.88–84.02	0.0006	UP
M2	QCR1	P31930	Cytochrome b-c1 complex subunit 1, mitochondrial	84	24	48	5.94–53.29	6.01–57.03	0.0018	DW
M4	ACTB	P60709	Actin, cytoplasmic 1	80	10	27	5.29–42.05	5.41–40.11	0.0008	UP
M5	CAZA1	P52907	F-actin-capping subunit alpha-1	76	9	38	5.45–33.03	5.95–33.74	0.0019	UP
M5bis	RUVB2	Q9Y230	RuvB-like 2	116	34	69	5.49–51.29	5.49–51.29	0.0024	UP
M8	CAPZB	P48637	F-actin-capping subunit beta	56	21	58	5.69–30.95	5.43–31.03	0.0005	UP
M9	CPNE1	Q99829	Copine-1	68	20	24	5.52–59.64	5.73–59.15	0.0001	UP
M10	PSME1	Q06323	Proteasome activator complex subunit 1	46	11	34	5.78–28.87	5.92–29.94	0.0041	UP
M11	PRDX4	Q13162	Peroxiredoxin-4	82	15	64	5.86–30.74	5.96–31.06	0.0007	UP
M16	GRP75	P38646	Stress-70 protein	26	7	17	5.87–73.92	5.88–77.11	0.0029	UP
M17	CPNE1	Q99829	Copine-1	103	18	24	5.52–59.64	5.68–60.01	0.0003	UP
M17b	2AAA	P30153	Serine/Threonine-protein phoshatase 2, 65 kDa regulatory subunit A alpha	91	23	38	5.00–66.06	5.11–65.91	0.0008	DW
M23	VIME	P08670	Vimentin	175	35	61	5.06–53.67	5.12–55.07	0.0022	DW
M42	CPNE1	Q99829	Copine-1	116	25	34	5.52–59.64	5.63–58.32	0.0012	UP
M319b	TPIS	Q02790	Triosephoshate isomerase	62	11	44	6.45–26.93	6.80–27.42	0.0015	DW

The proteins identified concern *HOMO SAPIENS.19453*
^(a)^: AC = accession number. ^(b)^: Score = 10*Log(*p*), where *p* corresponds to the probability that the observed match is a random event, according to the Swiss-Prot/NCBI database using the MASCOT search engine. ^(c)^: Sequence coverage = ratio between the sequence of the portion covered by matched peptides and the entire length of the protein sequence. pI/Mr: isoelectric point/molecular range. *p* value = statistical significance of the changes in the protein expression induced by GSC treatment with BzATP as compared to the expression of the same protein identified in EVs from control GSCs. Variation: indicates whether the proteins were UP- or DOWN-regulated. UM: unmatched (proteins), M: matched (proteins) in comparison to proteins identified in EXOs from control GSCs.

**Table 2 cells-13-00571-t002:** Matched proteins between MVs and EXOs from GSCs exposed to P2X7R stimulation.

SPOT ID	Abbr. Name	AC ^a^ Swiss/NCBI	Protein Description	Score ^b^	Peptide Matched	SC ^c^ %	Theoretical (pI/Mr)	*p*-Value	Variation
L7b	CSN4	Q9BT78	COP9 signalosome complex subunit 4	76	27	63	5.57–46.52	0.0001	UP
L14	PRDX2	P32119	Peroxiredoxin-2	74	8	42	5.29–42.05	0.0004	UP
L13	PRDX4	Q13162	Peroxiredoxin-4	63	11	43	5.86–30.74	0.0005	UP
L15	FRIL	P02792	Ferritin light chain	120	10	51	5.51–20.06	0.0021	UP
L22	PSA6	P60900	Proteasome subunit alpha type-6	59	11	44	5.06–53.67	0.0017	UP
L23	ATP23	Q9Y6H3	Mitochondrial inner membrane protease ATP homolog	56	5	23	5.11–66.65	0.0011	UP
L25	ARP3	P61158	Actin-related protein 3	88	17	33	5.61–47.79	0.0009	UP
L27	CLC1	P35523	Chloride intracellular channel protein 1	57	5	21	5.09–27.24	0.0024	DW

The proteins identified concern *HOMO SAPIENS.19453* ^(a)^: AC = accession number. ^(b)^: Score = 10*Log(*p*), where *p* corresponds to the probability that the observed match is a random event, according to the Swiss-Prot/NCBI database using the MASCOT search engine. ^(c)^: Sequence coverage = ratio between the sequence of the portion covered by matched peptides and the entire length of the protein sequence. pI/Mr: isoelectric point/molecular range. *p* value = statistical significance of changes in the protein expression induced by GSC treatment with BzATP as compared to the expression of the same protein identified in EVs from control GSCs. Variation: indicates whether the proteins were UP- or DOWN-regulated. L: indicates the proteins taken as a reference point for the alignment of the other proteins on the image of each gel.

**Table 3 cells-13-00571-t003:** Changes in the expression of proteins in CGS-derived EXOs induced by cell stimulation of P2X7Rs.

** *(a) Ex-Novo Induced Proteins by P2X7R Stimulation of GSCs* **										
**SPOT** **ID**	**Abbr.** **Name**	**AC ^a^ Swiss/NCBI**	**Protein Description**	**Score ^b^**	**Peptide** **Matched**	**SC** ^**c**^ **%**	**Theoretical** **(pI/Mr)**	**Experimental** **(pI/Mr)**	***p*-Value**	
UM6	VIME	B0YJC5	Vimentin	45	9	34	4.68–26.95	4.89–27.12	0.0004	
UM9	PSB9	P38646	Proteasome subunit beta type-9	68	10	39	4.93–23.36	5.03–23.78	0.0011	
UM15	PSA6	P60900	Proteasome subunit alpha type-6	66	11	42	6.34–27.83	6.44–28.03	0.0008	
UM21	PSA5	P28066	Proteasome subunit alpha type-5	57	9	38	4.74–26.56	4.55–26.71	0.0002	
UM92	PSA2	P25787	Proteasome subunit alpha type-2	58	6	29	6.92–25.99	6.98–26.11	0.0004	
UM105	EF1G	P26641	Elongation factor 1-gamma	78	10	24	6.25–50.42	6.25–52.01	0.0013	
UM110	RBBP4	Q09028	Histone binding protein RBBP4	57	13	31	5.03–53.51	5.13–53.89	0.0003	
** *(b) Changes in Top Protein Levels Caused by P2X7R Stimulation of GSCs* **										
**SPOT** **ID**	**Abbr.** **Name**	**AC ^a^ Swiss/NCBI**	**Protein Description**	**Score ^b^**	**Peptide** **Matched**	**SC** ^**c**^ **%**	**Theoretical** **(pI/Mr)**	**Experimental** **(pI/Mr)**	***p*-Value**	**Variation**
M7d	TIMP2	P16035	Metalloproteinase inhibitor 2	112	36	45	4.97–83.55	4.98–79.89	0.0006	UP
M38	MMP2	P08253	72KDa type IV collagenase	189	35	54	5.23–74.91	5.23–76.99	0.0015	UP
M41	MPP2+ GPSM2	Q14168 P81274	MAGUK p55 subfamily member 2; G-protein signaling modulator 2	47 66	7 23	13 29	6.32–64.82 5.97–76.61	6.32–65.12 6.10–77.11	0.0007	UP
M50	PDIA3	P30101	Protein disulfide-isomerase A3	72	12	38	5.98–57.14	5.96–56.78	0.0008	UP
M52	IF4A1	P60842	Eukaryotic initiation factor 4A	59	17	40	5.32–46.35	5.17–44.03	0.0005	UP
M54	ATPB	P06576	ATP synthase subunit beta, mitochondrial	87	12	31	5.26–56.52	5.32–56.90	0.0019	DW
M55	ENOG	P08670	Gamma enolase	57	9	38	4.91–47.58	4.91–49.01	0.0041	UP
M56	ARP3	P61158	Actin-related protein 3	84	19	38	5.61–47.79	5.46–49.15	0.0029	UP
M63	VIME	P08670	Vimentin	213	46	67	5.06–53.67	5.40–58.13	0.0008	DW
M63B	VIME	P08670	Vimentin	288	41	69	5.06–53.67	5.36–59.66	0.0004	UP
M65	VIME	P08670	Vimentin	115	31	54	5.06–53676	5.11–53.18	0.0022	UP
M69	SPB6	P35237	Serpin B6	74	14	50	5.18–42.93	4.79–44.03	0.0012	UP
M75	ACTB	P60709	Actin, cytoplasmic 1	46	6	15	5.29–42.05	5.46–47.12	0.0029	DW
M77	TPM4	P67936	Tropomyosin alpha-4 chain	60	18	64	4.67–28.61	4.80–30.11	0.0003	UP
M78	POC1	Q8NBT0	Procollagen C-endopeptidase enhancer 1	109	13	37	7.41–48.79	7.80–50.06	0.0008	UP
M92	TIMP2	P16035	Metalloproteinase inhibitor 2	98	12	44	7.45–25.06	7.48–25.34	0.0004	UP
M126	CBPE	P16870	Carboxypeptidase E	57	13	31	5.03–53.51	4.99–55.07	0.0022	UP
M138	ACTG	P63261	Actin, cytoplasmic 2	60	8	20	5.31–42.10	5.42–40.77	0.0012	UP

The proteins identified concern *HOMO SAPIENS.19453* ^(a)^: AC = accession number. ^(b)^: Score = 10*Log(*p*), where *p* corresponds to the probability that the observed match is a random event, according to the Swiss-Prot/NCBI database using the MASCOT search engine. ^(c)^: Sequence coverage = ratio between the sequence of the portion covered by matched peptides and the entire length of the protein sequence. pI/Mr: isoelectric point/molecular range. *p* value = statistical significance of changes in the protein expression induced by GSC treatment with BzATP as compared to the expression of the same protein identified in EVs from control GSCs. Variation: indicates whether the proteins were UP- or DOWN-regulated. UM: unmatched (proteins), M: matched (proteins) in comparison to proteins identified in EXOs from control GSCs.

**Table 4 cells-13-00571-t004:** Validation of protein sequences.

Label	ABBR. Name	Mw/p*I* Theor.	PMF Score ^a^	Peptide Matched/ Peptide Searched	SC ^b^ %	Lift (MS_2_) Ion Parent Masses (*m*/*z*)	Score ^c^ Tof-Tof	Peptide Sequence
M54	ATPB	5.26/ 56.52	87	12	31	2266.084 1815.869 1088.635	176	IPSAVGYQPTLATDMGTMQE R R.EVAFHGGIPDTGFYR.F VVDLLAPYAK
M92	TIMP2	7.45/ 25.06	98	12	44	1676.823 949.546 884.481	118	EVDSGNDIYGNPIKR RIQYEIK FFACIKR
M56	ARP3	5.61/ 47.79	84	19	38	2135.1205 1094.584 1041.5437	221	LGYAGNTEPQFIIPSCIAIK QYTGINAISK FMEQVIFK
M2	QCR1	5.94/ 53.29	84	24	48	1996.975 971.526	106	NALVSHLDGTTPVCEDIGR NRPGSALEK
M4	ACTB	5.29/ 42.05	80	10	27	2215.0699 1516.7026	189	DLYANTVLSGGTTMYPGIAD R QEYDESGPSIVHR
M8	CAPZB	5.69/ 30.95	56	21	58	1534.8298 1507.6879 901.4989	287	LTSTVMLWLQTNK SDQQLDCALDLMR NDLVEALK

^a^ PMF Score: values are Log_10_ (*p*), where *p* is probability that the observed match is a random event, as inferred by the Swiss Prot database using the MASCOT searching program; ^b^ SC: Sequence coverage means the ratio between the portion of the sequence covered by matched peptide and the full length of the protein sequence. ^c^ Score Tof-Tof: score that results from combining PMF and MS/MS matched peptide from ion parent fragments.

## Data Availability

Data are available upon request by qualified researchers to the corresponding author.

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
