# Peer review of "Changes Induced by P2X7 Receptor Stimulation of Human Glioblastoma Stem Cells in the Proteome of Extracellular Vesicles Isolated from Their Secretome"

_cells, 2024, doi:10.3390/cells13070571_

Round 1

Reviewer 1 Report

Comments and Suggestions for Authors

This revised manuscript has been improved by the authors. 

However, the authors have not addressed my first comment – independently validating a small number of the proteins identified in their Mass Spec experiments. I am aware antibodies are expensive, but I did not suggest that they validate every protein just a few of their most important hits. The ‘validation’ they have added is of the data they already presented and as such does not address my concern.

The additional Go enrichment analysis presented in new figure 3 is not informative at the level they have presented the data – i.e. catalytic or binding and as such does not add to the manuscript. What would be more informative if they had focused on function/pathway the proteins are involved in i.e cytoskeletal rearrangement, metabolism etc. Figures 4 and 5 however, are an excellent addition to the manuscript - The STRING analysis is more informative as the clusters can be linked to functions as the authors have done in the discussion.

The revised discussion is improved.

Author Response

However, the authors have not addressed my first comment – independently validating a small number of the proteins identified in their Mass Spec experiments. I am aware antibodies are expensive, but I did not suggest that they validate every protein just a few of their most important hits. The ‘validation’ they have added is of the data they already presented and as such does not address my concern.

R.: That WB analysis would have been not so expensive considering a limited number of proteins, we would like to underline that to obtain said proteins, it would have been necessary to: i) put a very large number of cells back into culture (we are in the order of billions); ii) perfom the extraction and separation of EVs (MVs and EXOs) from the culture medium; iii) extract a sufficient number of proteins from them to be used for WB analyses. Furthermore, it would have been necessary to make more gels, using antibodies not yet well known to our group. Overall, costs would not have been as limited as our funds unfortunately are. We hope that this reviewer understands our good will and at the same time our difficulties and we thank him for his/her patience.

We also want to emphasize that the two-dimensional electrophoresis we used allows a good separation of the proteins in a wide pH range and this remarkably reduces the possibility of contamination or evaluation errors. As well, by the LIFT technique we confirmed the protein identification obtained by the MS since by the LIFT procedure we identified a number of peptides which univocally correspond to a recognized protein. The techniques we used are based on MS but it comprises two different steps.

The additional Go enrichment analysis presented in new figure 3 is not informative at the level they have presented the data – i.e. catalytic or binding and as such does not add to the manuscript. What would be more informative if they had focused on function/pathway the proteins are involved in i.e cytoskeletal rearrangement, metabolism etc. Figures 4 and 5 however, are an excellent addition to the manuscript - The STRING analysis is more informative as the clusters can be linked to functions as the authors have done in the discussion.

R.: we followed the reviewer’s advice and changed the Figure 3, leaving the panel with pies regarding the biological processes (related to protein functions in cell processes) and reintroducing the pies originally included about the Pathways activated by these proteins. Accordingly, we modified the Table S1-3, included in the Supplemental material, containing the detailed description of the function and pathways of each investigated protein

Reviewer 2 Report

Comments and Suggestions for Authors

This study provides a new understanding for the role of P2X7R in GBM aggressiveness through EVs. Making P2X7R a new potential therapeutic target for human GBM.

Author Response

we thank this Reviewer